# Ultra-narrow inhomogeneous spectral distribution of telecom-wavelength vanadium centres in isotopically-enriched silicon carbide

Pasquale Cilibrizzi[1], Muhammad Junaid Arshad [1], Benedikt Tissot [2], Nguyen Tien Son [3], Ivan G. Ivanov[3], Thomas Astner [4], Philipp Koller [4], Misagh Ghezellou [3], Jawad Ul-Hassan [3], Daniel White[1], Christiaan Bekker[1], Guido Burkard [2], Michael Trupke [4] ✉ & Cristian Bonato [1] ✉

Spin-active quantum emitters have emerged as a leading platform for quantum technologies. However, one of their major limitations is the large spread in optical emission frequencies, which typically extends over tens of GHz. Here, we investigate single $V^{4+}$ vanadium centres in 4H-SiC, which feature telecom-wavelength emission and a coherent $S = 1/2$ spin state. We perform spectroscopy on single emitters and report the observation of spin-dependent optical transitions, a key requirement for spin-photon interfaces. By engineering the isotopic composition of the SiC matrix, we reduce the inhomogeneous spectral distribution of different emitters down to 100 MHz, significantly smaller than any other single quantum emitter. Additionally, we tailor the dopant concentration to stabilise the telecom-wavelength $V^{4+}$ charge state, thereby extending its lifetime by at least two orders of magnitude. These results bolster the prospects for single V emitters in SiC as material nodes in scalable telecom quantum networks.

The demonstration of long-distance quantum networks represents a new paradigm for communication security[1]. Single optically-active spin defects and impurities are strong candidates for physical nodes in such systems[2–4], and have been used in many of the most successful implementations of quantum networking primitives to date. A general architecture of quantum networks consists of an electronic spin associated with a point defect or impurity, interfaced to optical photons through spectrally stable spin-selective optical transitions. The electronic spin can couple to one or more nuclear spins[5–7], which can serve as long-term quantum state storage sites and as ancillary qubits for implementing quantum error correction[8,9]. Examples of physical

systems used to implement this architecture include the nitrogen-vacancy[10,11], silicon-vacancy[12,13] and tin vacancy[14] centres in diamond; related systems in SiC[15–19], silicon[20–23] and 2D materials[24]; and rare-earth ions in crystals[25–28].

Quantum network applications can benefit from spin-photon interfaces in the telecom wavelength region, which are inherently compatible with standard optical fibre networks. While emission at alternative wavelengths can be converted to the telecom range by nonlinear optical processes[29–31], the additional hardware requirement reduces the overall system efficiency and can add noise. The spin-photon interface needs to be near lifetime-limited and spectrally stable

[1]School of Engineering and Physical Sciences, SUPA, Heriot-Watt University, Edinburgh EH14 4AS, United Kingdom. [2]Department of Physics, University of Konstanz, D-78457 Konstanz, Germany. [3]Department of Physics, Chemistry and Biology, Linköping University, SE-581 83 Linköping, Sweden. [4]Institute for Quantum Optics and Quantum Information (IQOQI), Austrian Academy of Sciences, A-1090 Vienna, Austria. ✉e-mail: michael.trupke@oeaw.ac.at; c.bonato@hw.ac.uk

to ensure high-visibility quantum interference. Additionally, to entangle different nodes in a quantum network, the inhomogeneous distribution of the optical transition frequencies for different emitters must be small to ensure indistinguishability of the emission, ideally close to the linewidth of individual defects.

Quantum emitters typically exhibit a spread in emission frequencies induced by local differences in strain and electric field within the solid-state matrix. This problem has been addressed by tuning the optical transitions through the external application of either strain or electric fields[32,33]. Scaling up to multiple qubits is not trivial, however, due to the technical complexity of adding more and more electrical contacts or piezoelectric elements, while minimising cross-talk. An alternative solution is to exploit the same nonlinear process used to convert single photons to the telecom range, tuning the frequency of the pump laser to finely control the output telecom frequency so that all photons from all emitters are brought on resonance[30]. This approach, however, adds substantial technological complexity for each emitter in the network.

Vanadium centres in silicon carbide (SiC) have recently received considerable interest for use in quantum networks, due to several attractive properties. Their emission directly in the O-band telecom range (1260 nm to 1360 nm)[34,35] is directly compatible with standard fibres used in commercial optical telecommunications networks[36] and allows interfacing with low-loss photonic circuitry[4,37]. Direct emission in the telecom O-band removes the need for complex wavelength conversion and tuning hardware, and unlocks the possibility to wavelength-multiplex the quantum signal in the same standard telecom fibre with the classical signal in the C-band. Recent work has demonstrated long electron spin relaxation times ($T_1$) of up to 25 s[38], ensemble dephasing times ($T_2^*$) of several microseconds and ensemble Hahn echo $T_2$ well in excess of 100 μs[39] at cryogenic temperatures. SiC as a host material is widely used in high-power electronics, with established recipes for industrial-scale growth, doping, and fabrication, with very promising linear and nonlinear parameters for photonics[40]. However, despite these encouraging features, several questions related to its suitability for application in a spin-photon interface remain open. In particular, the inhomogeneous spectral broadening in ensembles and single emitters is large compared to the natural linewidth of the defects, and stable emission of the neutral charge state in single defects has so far only been observed when applying a repump ultraviolet (UV) laser[35].

Here, we investigate the optical and electronic properties of single neutral vanadium (V) centres in the 4H polytype of SiC (4H-SiC), providing experimental verification of spin-conserving optical transitions and their dependence on the applied magnetic field. We systematically study the optical emission of hundreds of V centres, comparing their distributions in standard and isotopically-enriched SiC. We find an ultra-narrow ~100 MHz inhomogeneous spectral distribution in isotopically enriched SiC, compared to several GHz in SiC with a natural abundance of isotopes. This distribution is significantly smaller than any other single quantum emitter previously reported in the literature[21,23,41–47]. We trace the origin of the ultra-narrow distribution to the reduction of local stress in the isotopically enriched SiC, resulting from the nearly complete removal of any spread in the mass of elements in the crystal, i.e. the prevalence of different isotopes. Finally, we investigate the charge state dynamics of vanadium centres in SiC, demonstrating that, by tailoring the material purity, we can stabilise the required charge state for at least several seconds. Our results reveal the great potential of single V centres in SiC for scalable telecom quantum networking.

## Results

### Spin-selective optical transitions

A V centre forms when a V atom substitutes a silicon atom in the SiC crystal lattice[34,35,48]. In 4H-SiC, the neutral ($V^{4+}$) state is the only charge state that exhibits luminescence in the telecom region, featuring two zero-phonon lines (ZPL), denoted as $\alpha$ and $\beta$[34], and an electronic spin $S = 1/2$.

We identify $V^{4+}$ centres by confocal spectroscopy at 4.3 K (see Methods and Supplementary Note 1), using a narrowband tuneable CW laser (1278.8 nm) to resonantly excite the $\alpha$ zero-phonon line (ZPL) of $V^{4+}$[34], while detecting the phonon sideband emission (1300 nm - 1600 nm). The typical excitation powers range from 1 μW to 4 μW depending on the specific experiments. We use a green repump laser (520 nm, 14 μW) to compensate for laser-induced ionisation[35]. The necessity of a repump laser is discussed in more detail in the section concerning the stability of the neutral charge state and in Supplementary Note 2. By scanning the excitation lasers across an isotopically-enriched sample (sample A, described in Methods), we obtain bright photoluminescence (PL) spots as shown in Fig. 1a. An automated detection algorithm (see Methods) finds 389 spots in Fig. 1a, corresponding to 0.29 spots/μm² over a map area of 1600 μm². We confirm the single emitter nature of one of the spots by $g^{(2)}(\tau)$ auto-correlation measurements, observing a $g^{(2)}(0) = 0.255 \pm 0.180$ at zero delay (Fig. 1b).

We investigate the single $V^{4+}$ centre through photoluminescence excitation (PLE) spectroscopy, by scanning the telecom laser frequency across the $\alpha$ ZPL with selected $\sigma_+$ ($\sigma_-$) circular polarisation at a magnetic field of 1000 Gauss, oriented along the SiC c-axis (see Methods and SI for details). The PLE measurements reveal two distinct peaks with orthogonal circular polarisations (Fig. 1c). We fit the PLE spectra with Lorentzian functions, centred at $0.155 \pm 0.015$ GHz ($\sigma_+$) and $-0.335 \pm 0.011$ GHz ($\sigma_-$). The relative Zeeman splitting of 510 MHz, extracted from the fittings, is in agreement with theoretical predictions[49,50] (see Supplementary Note 3 for details). The $\sigma_+$ and $\sigma_-$ peaks in Fig. 1c, result from circular polarisation-dependent selection rules[50] and are associated with the spin-conserving transitions between the first ground ($GS_1$) and first excited state ($ES_1$), as schematically shown in Fig. 1d. The electronic structure of $V^{4+}$ is determined by a single active electron localised in a $d$ orbital, which possesses $C_{3v}$ symmetry imposed by the SiC crystal field. The orbital degeneracy is lifted by spin-orbit coupling into Kramers doublets (KDs): pairs of degenerate states connected through time inversion. The degeneracy of the Kramers doublets can be lifted by an applied magnetic field[49–51] (see Supplementary Note 3 for details). As a consequence, spin-selective optical transitions between KDs in the ground and excited states addressable by circularly polarised excitation light[49] are expected.

We further investigate the magnetic field dependence of the optical transitions through polarisation-resolved ($\sigma_+$ and $\sigma_-$) PLE spectroscopy as a function of the magnetic field strength $B$ applied along the c-axis (Fig. 2).

In the absence of an external magnetic field (B = 0 Gauss), the two $\sigma_+$ and $\sigma_-$ circularly polarised PLE spectra are superposed with no visible splitting. At B = 600 Gauss, a splitting of 220 MHz becomes visible, which increases to 510 MHz at B = 1000 Gauss. Remarkably, the linewidth of the resonance decreases for a higher applied magnetic field. We compare our data with the theoretical models (Fig. 2d) based on the $d$-orbital configuration and $C_{3v}$-symmetry of the centre[49,50,52], calculating the relative frequencies and strengths of the hyperfine transitions using previously measured experimental values[35,38,52] (adapted to the symmetry-based hyperfine model[50], see Supplementary Note 3). We assume a Lorentzian shape with the same full-width $\gamma$ for all hyperfine-allowed transitions. The width $\gamma \approx 1$ GHz, the defect-dependent central (or zero-field) transition frequency $\Delta_{cr}$, as well as parameters describing the background offset and the amplitude of the peak are fit parameters. Further details on the fit and model are presented in Supplementary Note 3.

In simple terms, the decrease in linewidth with increasing magnetic field can be understood from the fact that the off-diagonal coupling, which is of a vastly different form for the involved KDs, is

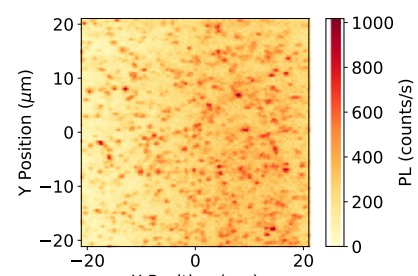

(a) photoluminescence map of vanadium emitters

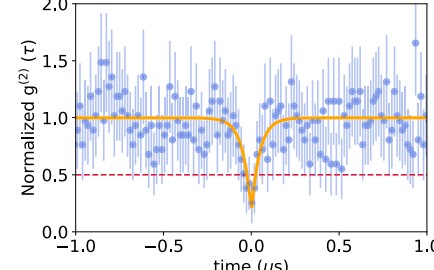

(b) single vanadium centre

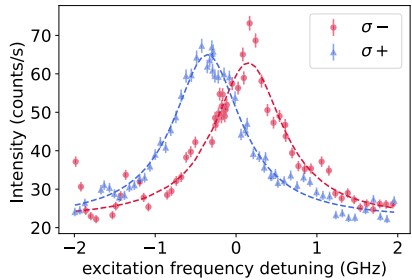

(c) polarisation-resolved PLE spectroscopy

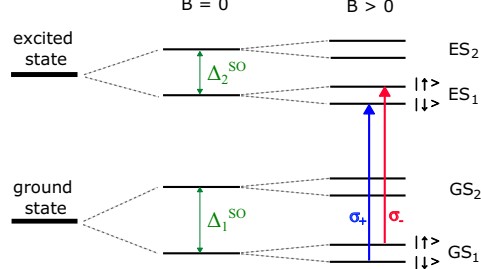

(d) simplified energy level scheme

**Fig. 1 | Spectroscopy of single V$^{4+}$ centres in 4H-SiC. a** Photoluminescence (PL) intensity map acquired by scanning the telecom excitation laser (2.2 μW) across the sample on resonance with the α ZPL of the V centres (1278.84 nm, 234.42457 THz). The V centres appear as localised PL spots. A 14 μW repump is applied during the scan. **b** Second-order autocorrelation measurement g$^{(2)}$(τ) for one of the spots, as a function of time delay (τ). The data are not corrected for background counts. The measured g$^{(2)}$(0) = 0.255 shows that the spot corresponds to a single V centre. The yellow line corresponds to a single exponential fit function, f(x) = 1 - A e$^{-|t|/t_1}$, with A = 0.828 ± 0.163 and t$_1$ = 0.048 ± 0.013 μs. The decay constant here is different than the known value of 167 ns for the emitter lifetime[35], as it depends on the excitation optical power[70]. **c** Polarisation-resolved photoluminescence excitation (PLE) measurements, performed at an applied magnetic field of 1000 Gauss. For

these measurements, we use excitation powers of 14 μW for the repump laser and 2.2 μW for the telecom laser. The integration time is set at 20 seconds per data point, with each point acquired three times and then averaged. The red circles and blue triangles correspond respectively to σ$_-$ and σ$_+$ circular polarisation. The frequency of the laser is expressed as detuning from the base frequency f$_0$ = 234425 GHz. **d** Simplified energy level diagram for V centres in SiC. The degeneracy of the ground (GS) and excited (ES) states, set by the C$_{3v}$ symmetry, are broken by spin-orbit coupling (Δ$_1^{SO}$ and Δ$_2^{SO}$)[49,50]. An applied magnetic field B further removes the degeneracy between the electron spin levels. The blue and red arrows indicate the excitation of the two spin-dependent transitions with circularly polarised light (σ$_+$ and σ$_-$, respectively). The error bars in (**b**) and (**c**) correspond to the Poisson noise on the photon counts.

---

suppressed under large magnetic fields and, if the *zz* components share the same sign, the diagonal coupling leads to a similar hyperfine splitting. Therefore, the model (using the literature values for the hyperfine parameters of "bulk" defects much deeper than the size of their wavefunctions[53], with intact symmetry) predicts that the electron-spin-conserving transitions will converge at higher magnetic fields, resulting in a narrowing of the linewidth. The splitting of the two peaks can be approximated by $\mu_B|g_e - g_g|B$ for large magnetic fields $B \gg 100$ Gauss, in agreement with our experimental measurement.

## Ultra-narrow inhomogeneous spectral distribution

We next investigate the spectral inhomogeneous distribution of the ZPLs corresponding to different V$^{4+}$ centres. Previous experiments on V ensembles[35] have highlighted an asymmetry in the ZPL, with a longer tail and duplicated lines at higher frequencies. This has been attributed to the isotope shift from nearby $^{29}$Si, $^{30}$Si and $^{13}$C isotopes. The isotope shift is a change in frequency of the ZPL given by a variation in the mass of nearby atoms, corresponding to heavier or lighter isotopes, that changes the local strain or bandgap[54–56]. Linewidth fitting of ensemble optical spectroscopy data[35] has given a shift of 22 ± 3 GHz per unit mass for nearest-neighbour carbon isotopes and 2.0 ± 0.5 GHz per unit mass for silicon isotopes. Further evidence of this was recently provided by optically-detected magnetic resonance (ODMR) measurements[39], showing a change in the ODMR spectrum as the optical excitation wavelength of a V ensemble was tuned. In particular, while excitation at a wavelength of 1278.86 nm (a detuning of about -3 GHz compared to our base frequency f$_0$) only showed a dip related to the V centre electron spin resonance, excitation at 1278.76 nm (about +15 GHz from

f$_0$) evidenced the presence of side peaks consistent with hyperfine coupling to neighbouring $^{29}$Si atomic nuclei. This suggests that when SiC is isotopically enriched, removing most $^{29}$Si, $^{30}$Si and $^{13}$C isotopes, the inhomogeneous distribution of the ZPL should become narrower compared to the case of isotope composition according to natural abundance.

We investigate this quantitatively by performing a sequence of PLE maps, each at different excitation frequency, in two different SiC samples, one isotopically enriched (sample A), and one featuring a natural abundance of Si and C isotopes (sample B). Both samples are described in detail in the Methods. A sub-set of the maps is shown in Fig. 3a, with the full sequence reported in the Supplementary Note 4B. To quantify the spectral distribution of the central frequency of the V emission, we perform a statistical analysis of the PLE spots in the maps. We process the series of maps by automatically detecting spots of sizes compatible with the diffraction limit, and fitting the PLE intensity for each spot, as a function of excitation frequency, with a Gaussian function to extract the centre frequency (details in the Supplementary Note 4). Histograms for the inhomogeneous distributions of central frequencies in the two samples are shown in Fig. 3c. In sample B, with a natural abundance of isotopes, the frequencies span across several GHz, with a few peaks in the distribution, compatible with the ~ 2 GHz per unit mass expected for Si isotopes[35]. In future work, it will be interesting to correlate optical spectroscopy with optically-detected electron spin resonance measurements to investigate the hyperfine coupling for each different detuning of the ZPL. This was not possible here, since we could not detect any ODMR on single centres, likely due to the short electron spin relaxation timescales at 4 K for V[38]. In the

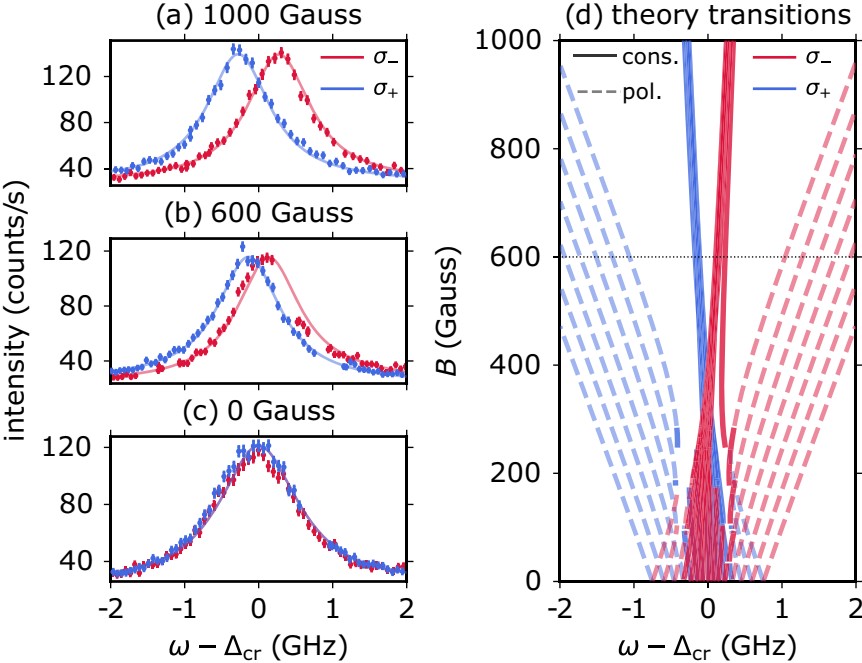

**Fig. 2 | PLE spectroscopy as a function of the applied magnetic field. a–c** PLE spectroscopy of the same PL spot, with selected $\sigma_-$ (red) and $\sigma_+$ (blue) circular polarisation excitation, under different magnetic fields (0 Gauss, 600 Gauss and 1000 Gauss, respectively) applied along the sample c-axis. For this experiment, we do not confirm the single nature of the emitters with $g^{(2)}(\tau)$ measurements. Here we use 14 µW for the repump laser and 1 µW for the telecom laser. The integration time is 15 seconds per data point, and each point is acquired three times and subsequently averaged. The error bars correspond to the Poisson noise of the photon counts. **d** Theoretical spin-conserving (cons.) and nuclear spin-polarising (pol.) hyperfine allowed transitions predicted by the model outlined in Supplementary

Note 3 and using hyperfine parameters from the literature[35,38,52]. The electron-spin-conserving transitions between the $ES_1$ and $GS_1$ (Fig. 1d) electron levels (solid lines) narrow for higher magnetic fields and conserve the nuclear and Kramsers doublet (KD) spin, while the hyperfine allowed transitions (dashed lines) flip the KD spin and can polarise the nuclear spin. The fit (solid lines in (**a–c**) allows for different amplitudes and backgrounds for each of the measurements, but shares the same width for all transition frequencies (see **d**) 1038 ± 7 MHz and central transition frequency $\Delta_{cr}$ = 234425594 ± 4 MHz. The polarisation information is encoded in the curve colour (see legend).

isotopically-enriched sample, the central frequencies exhibit a much narrower distribution, with standard deviation -100 MHz. Over three regions in sample A, the central frequencies of the distributions are all quite close: $f_A$ = 227 ± 105 MHz, $f_B$ = -41 ± 109 MHz, $f_C$ = -25 ± 76 MHz, comparable to the absolute accuracy of the wavemeter (-150 MHz). A fourth measurement in a different region of the crystal shows a detuning of 700 MHz. This measurement was taken after a thermal cycle of warm-up and cool-down, which may have affected the sample strain. However, within each region, we observe the same 100 MHz ultra-narrow spectral distribution (see Supplementary Note 4B).

We can trace the origin of the frequency shifts in the optical transitions of $V^{4+}$ centres to the local strain produced by nearby isotopes[35]. Theoretically, we will treat the coupling to strain analogously to the coupling to electric fields[49]. This is possible for several reasons: the symmetry arguments were used on the orbital level, the fact that the strain coupling fulfils time-reversal symmetry and the possibility to assign (combined) strain tensor elements to the same irreducible representations as the electric field components[57]. In particular, there are two strain tensor elements (i.e., $\epsilon_{zz}$ and $\epsilon_{xx} + \epsilon_{yy}$) that transform like an electric field in the $z$-direction within the $C_{3v}$ symmetry which can directly influence the energy spacing of the KDs[58].

A local inhomogeneous distribution on the order of 100 MHz compares very favourably with other quantum emitters. This is several orders of magnitudes smaller than literature values for single nitrogen-vacancy (NV) centres in diamond ( - 40-50 GHz)[41]. A narrower inhomogeneous distribution, on the order of 10-20 GHz, has been reported for a single silicon-vacancy centre (SiV) in diamond[42,43] and silicon carbide ($V_{Si}$)[44]. V in SiC compares favourably even to single rare-earth ions and single emitters in silicon, which can feature narrow distributions down to the MHz-GHz level[28,46,59,60]. A more informative figure of

merit could be the ratio $\eta$ between inhomogeneous broadening and the transform-limited linewidth, which quantifies by how many line-widths the frequencies of two emitters need to be shifted to be brought into resonance. According to this metric (Supplementary Table 3), V in SiC performs on par with the SiV centre in diamond ($\eta$ - 100), a system protected against electric fields and strain by inversion symmetry, more than an order of magnitude better than the NV centre in diamond ($\eta$ - 3400).

### Stability of the neutral charge state

Finally, we investigate the stability of the neutral $V^{4+}$ charge state. In 4H-SiC, V can exist in three possible charge states within the bandgap: $V^{3+}$ (negatively charged), $V^{4+}$ (neutral) and $V^{5+}$ (positively charged)[61], with only the $V^{4+}$ state featuring optical emission in the telecom region and a $S$ = 1/2 electronic spin. The lifetime of the $V^{4+}$ charge state is important as it sets a limit for the use of V centres in quantum technology applications.

The stabilisation of a given charge state for a deep-level defect depends on the local Fermi level and the complex interplay with other nearby dopants/defects that act as electron donors or acceptors. The charge state transition levels for V and the other main defects in SiC, such as the carbon vacancy ($V_C$) and the divacancy ($V_{Si}V_C$), are displayed in Fig. 4a. We include a discussion in Supplementary Note 2 about how these levels have been determined in the literature.

As seen in previous experiments[35,38], resonant excitation of the -1278 nm line resulted in luminescence quenching due to the change in the charge state of $V^{4+}$, which is restored by a non-resonant (UV or green) repump laser. Here we characterise the charge dynamics of the $V^{4+}$ state though two experiments, each performed on both samples A and B, which feature two different nitrogen and boron doping

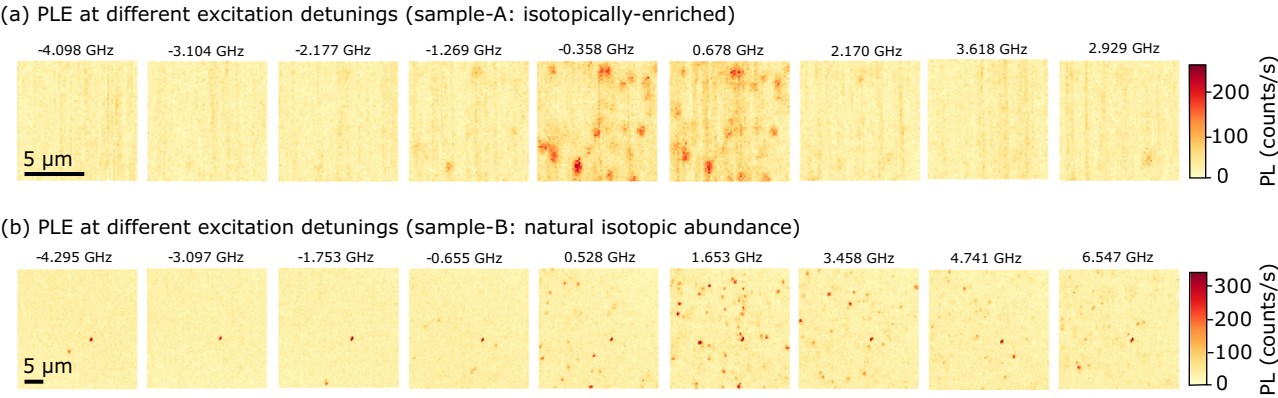

(a) PLE at different excitation detunings (sample-A: isotopically-enriched)

-4.098 GHz  -3.104 GHz  -2.177 GHz  -1.269 GHz  -0.358 GHz  0.678 GHz  2.170 GHz  3.618 GHz  2.929 GHz

5 µm

(b) PLE at different excitation detunings (sample-B: natural isotopic abundance)

-4.295 GHz  -3.097 GHz  -1.753 GHz  -0.655 GHz  0.528 GHz  1.653 GHz  3.458 GHz  4.741 GHz  6.547 GHz

5 µm

(c) histograms of central frequencies for PLE spectral peaks

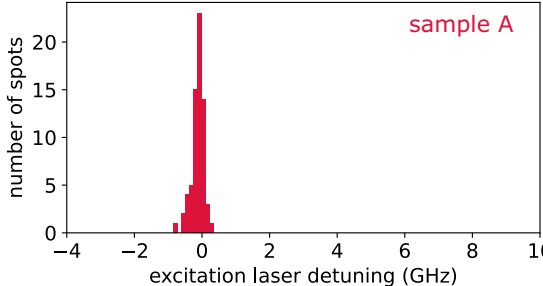
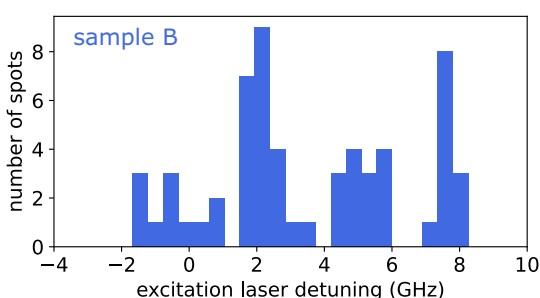

**Fig. 3 | Inhomogeneous spectral distribution of the V centres in SiC with different isotopical composition. a, b** Sequence of PLE maps at different detunings of the excitation laser (14 µW for the repump laser, 4 µW for the telecom laser), respectively for the isotopically-enriched sample (A) and the natural-abundance sample (B). In the isotopically-enriched sample the vast majority of emitters appear only in a narrow frequency range. Maps in (**a**) are 10 µm × 10 µm, maps in (**b**) 30 µm × 30 µm. We acquired larger maps for sample B because the concentration of spots in each map is smaller than that for sample A (due to the larger frequency spread). **c** Histograms for the central frequencies of 181 PL spots associated to V centres in the natural-abundance (right) and 61 spots in isotopically-enriched (left) samples. In the sample with a natural abundance of Si and C isotopes, the $V^{4+}$ centres are spectrally spread over several GHz, presenting a distribution with multiple peaks. In contrast, the distribution is much narrower in the isotopically-enriched sample, with a standard deviation of about 100 MHz.

concentrations (described in detail in the Methods). For sample A, we expect the Fermi level to be pinned at the N shallow donor level, while for sample B (semi-insulating) it is situated approximately in the middle of the bandgap (as shown in Fig. 4a).

In the first experiment (Fig. 4b), we probe the ionisation of the $V^{4+}$ centre due to the telecom (resonant) excitation laser, with the scheme shown in the inset of Fig. 4b. We prepare the $V^{4+}$ state with a green laser pulse (300 ms), and detect the PL decay at different CW telecom laser powers. We fit the decays at different excitation powers with a single exponential, and plot the decay rates, extracted from the previous fits (shown in Supplementary Note 2), as a function of the resonant excitation power (Fig. 4b). In both samples, the resonant excitation laser induces quenching of PL from $V^{4+}$, but the effect is more pronounced in sample A.

In the second experiment, we assess the lifetime of the neutral $V^{4+}$ charge state when no optical excitation is present (Fig. 4c). This is relevant, for example, for quantum memory experiments, when optical excitation is only used for short spin initialisation and readout, and the spin preserves a quantum state in the dark. To determine this timescale, we initialise the $V^{4+}$ state with a green laser pulse (300 ms) and subsequently apply a readout pulse with the resonant laser (probe, 300 ms), after a delay $\tau$ (see inset in Fig. 4c). By increasing the delay $\tau$ and monitoring the change in PL during the readout pulse, we measure the PL decay at different delays. We then fit the decays at different delays ($\tau$) using a single exponential, and plot the peak intensities extracted from the fits as a function of the delay time $\tau$ (Fig. 4c). For sample B in Fig. 4c, we repeated the experiment twice, and plotted the average of the two measurements. We observe that, even in the absence of laser excitation, the PL peak intensity decays exponentially

in sample A, with a decay constant of $129 \pm 6$ ms (violet curve). For sample B, this timescale is considerably extended, up to several seconds (orange curve). Note that this is a lower bound as, at these timescales, the decay is likely determined by unwanted excitation by the telecom laser leaking through the acousto-optic modulator when turned off, due to a non-ideal extinction ratio.

These experimental observations can be understood as being due to the different Fermi level in sample B compared to sample A, which improves the stability of the $V^{4+}$ charge state in sample B. To enable discrimination of single vanadium emitters, we used an implantation dose of $1 \times 10^8$ cm$^{-2}$ in both samples (see Methods). At this low dose, the Fermi level is not affected by V dopants but rather by the concentration of residual nitrogen (N) and boron (B), and other intrinsic defects such as divacancies and carbon vacancies (see Methods for details). Specifically, in sample A, the Fermi level is pinned to the donor level of N (see Fig. 4a "$E_F$, sample A"). In such n-type sample, V is expected to exist in the negative charge state $V^{3+}$ at equilibrium.

The re-pump laser induces excitation of electrons from the acceptor levels of V (0|-) to the conduction band, thus enabling charge state conversion from $V^{3+}$ (dark) to $V^{4+}$ (bright). The decay that we observe in Fig. 4c can be explained as an effect of the resonant laser used to probe $V^{4+}$, leading to transformation of $V^{4+}$ into $V^{3+}$. Indeed, the telecom resonant excitation pumps electrons from the shallow N donor and $V_C$ acceptor levels into the conduction band, which can be re-captured by $V^{4+}$ to create $V^{3+}$, leading to the measured exponential decays of its PL on a 129 ms timescale. We notice that the resonant excitation alone is most likely insufficient to reactivate $V^{4+}$, which stipulates the need of repump laser of higher energy (see Supplementary Note 2B).

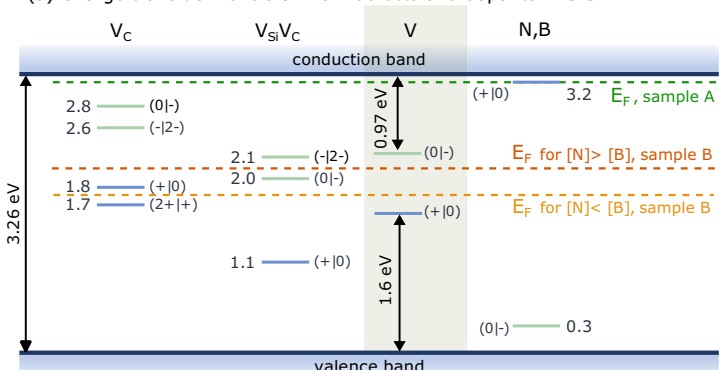

(a) Charge transition levels of main defects and dopants in SiC

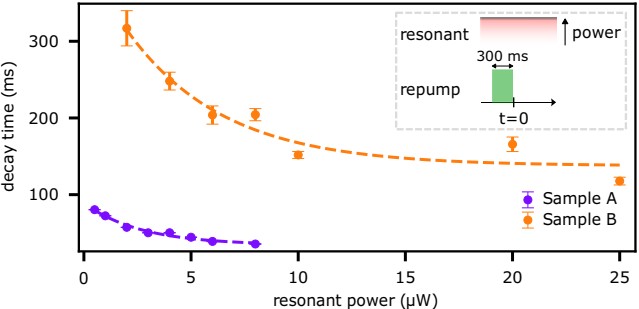

(b) Probing the ionization of the V$^{4+}$ centers

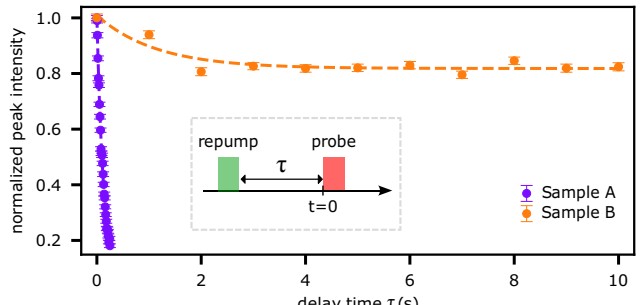

(c) Probing the lifetime of the neutral V$^{4+}$ charge state

**Fig. 4 | Stability of the V$^{4+}$ charge state. a** Energy diagram showing the charge transition levels for V centres, the main dopants nitrogen (N) and boron (B), and intrinsic defects, such as carbon vacancy (V$_C$) and divacancy, (V$_{Si}$V$_C$) involved in the V charge state dynamics in SiC. A transition between the positively-charged and neutral states, for example, is labelled as (+/0). V$^{4+}$ corresponds to the neutral (0) and V$^{3+}$ to the negative (-) charge state. In sample A (n-type), the N shallow donor can compensate the acceptor levels of shallow B and V$_C$, pinning the Fermi level (E$_F$) at the N shallow donor level and stabilising V$^{3+}$. In sample B (semi-insulating), the E$_F$ is located at the (0|−) acceptor level of the divacancy if the concentration of the N donor is larger than that of the B acceptor or, vice versa, at the ( + |0) donor level of V$_C$. In both cases, V$^{4+}$ is stabilised at equilibrium. **b** Decay of the V$^{4+}$ charge state at increasing powers of the resonant CW excitation. V$^{4+}$ is prepared by a 300 ms

repump pulse and the PL decay over time is recorded as a function of the resonant laser power (see pulse sequence inset). The ionisation induced by the resonant excitation follows an exponential decay of the count rate for both samples (see Supplementary Note 2), with a weaker decay in sample B, where the Fermi level stabilises V$^{4+}$. **c** Lifetime of the V$^{4+}$ charge state when no optical excitation is present. The system is prepared in the V$^{4+}$ state by a repump pulse and the PL emission intensity is then recorded as a function of the delay ($\tau$) between the repump and the probe. The PL peak intensity, corresponding to the occupation of the V$^{4+}$ charge state, is plotted against the delay $\tau$ for both samples (see Supplementary Note 2). In sample A, the V$^{4+}$ charge state decays rapidly with a timescale of 129 ± 6 ms, while it survives for several seconds in sample B. Errors bars in (**a**) and (**b**) correspond to the standard deviations extracted from the fits.

Furthermore, the higher the resonant laser power, the higher the excitation rate of free electrons, leading to higher free-electron concentration and increased capture rate to the V$^{4+}$ centres. This explains the observed decrease in the decay time observed in Fig. 4b.

In sample B, the residual N donors and B acceptors have comparable concentrations, in the low 10$^{15}$ cm$^{-3}$ range, and the carbon vacancy and divacancy are the dominant intrinsic defects (with a concentration in the high 10$^{15}$ cm$^{-3}$ range). The insulating character of this sample sets the Fermi level to the middle of the gap, stabilising V$^{4+}$. A more detailed discussion about the expected charge state dynamics at the microscopic level can be found in Supplementary Note 2.

## Discussion

In this work, we have investigated several open questions related to the spin-active V centre in SiC. We have utilised two types of SiC, one with isotopically-enriched composition but slightly n-doped, and one high-purity, semi-insulating sample with natural isotopic composition. We have provided experimental verification of spin-dependent optical transitions in V, investigating their dependence on the applied magnetic field and confirming recent theoretical predictions on its electronic structure[49,50]. Remarkably, we observe an ultra-narrow ~ 100 MHz spectral inhomogeneous distribution in the isotopically enriched sample: the comparison in the Supplementary Table 3 highlights that this is the narrowest observed for any single solid-state emitter. This is achieved through isotopic enrichment of the

SiC host which removes the isotope shift effect, with different isotopes locally inducing a different frequency shift in the emission[35]. A narrow inhomogeneous distribution is crucial to create remote entanglement between different emitters in the nodes of a quantum network, as high-visibility quantum interference requires perfect indistinguishability between the photons. It could also be beneficial for super-radiance and collective emission[62,63] experiments, that similarly require indistinguishability. This property could also be used to pre-select emitters with a specific configuration of nearby nuclear spins by optical spectroscopy, facilitating the implementation of specific emitters in a quantum memory. Finally, we engineer the material doping level to stabilise the V$^{4+}$ charge state, which features telecom emission and a spin $S = 1/2$ state. Doping level and isotopic composition are independent parameters, that can be combined to simultaneously achieve stabilisation of the correct charge state and narrow inhomogeneous distribution on the same sample.

One open question is related to the linewidth of the observed optical transitions. We measure a linewidth of about 600 MHz at the lowest excitation power, in contrast to a lifetime limit of less than 1 MHz. The electronic optical transition is broadened with respect to the lifetime limit by the presence of 16 hyperfine levels corresponding to the interaction with the $I = 7/2$ $^{51}$V nuclear spin, with a hyperfine interaction on the order of 200 MHz (see Supplementary Table 3). Including the hyperfine transitions, however, should result in a set of multiple separated MHz-linewidth lines, that would enable direct optical access

to the nuclear spin. Temperature can also affect the linewidth, through phonon-assisted dephasing. In our setup, experiments are limited to about 4 K and additional measurements at lower temperature are required to fully assess the impact of phonon dephasing. A further mechanism could be related to spectral diffusion induced by fluctuating electric fields associated with other defects in the vicinity of the V centres. The implantation process can result in lattice damage, especially for heavier atomic species like V. While we have annealed both samples at a very high temperature after implantation (see Methods), this may have not fully repaired the SiC crystal lattice. A possible way to assess and address the contribution of this factor may be to embed the vanadium centres in a p-i-n diode structure and apply a large electric field to empty all of the charge traps, stabilising the local electric field environment[64].

Single V centres can be detected without any photonic structure, in contrast, for example, to most rare-earth ion emitters. Through resonant excitation, we observe count rates on the order of 150 counts per second in the phonon sideband. With a Debye-Waller factor of about 0.3[34], this corresponds to about 75 counts per second in the ZPL. The low count rate is not completely explained by the relatively long excited-state lifetime (167 ns[35]), suggesting a reduced quantum efficiency of the emission. Given the rich hyperfine structure of $I = 7/2$ $^{51}$V nuclear spins, resonant optical excitation could lead to pumping into a dark state. If this is the case, nuclear spin polarisation[50] would boost counts and additionally provide access to a fast quantum memory, with hyperfine interaction on the order of 200 MHz. Optical collection could be enhanced through the use of photonic structures such as solid immersion lenses[65] or nanopillars[40]. Resonant photonic structures could further enhance emission: recent work on rare-earth ions has shown that Purcell factors exceeding few hundreds can be achieved both by photonic crystals[26], open microcavities[27] and plasmonic waveguides[66]. If the spin-photon interface is implemented exploiting the circular polarisation of the optical transition, it is important to preserve the polarisation degeneracy of the cavity modes, which requires controlling shape and birefringence in the case of micropillars or open cavities[67], or utilising symmetric geometries in the case of photonic crystals[68].

In conclusion, our results, in combination with recent measurements of long spin relaxation and coherence lifetimes below T=2 K[39], reveal the potential of single V centres in SiC for quantum networking and as sensitive probes of their crystalline environment. By engineering the isotopic composition and purity of the SiC host material, we have shown that it is possible to stabilise the $S = 1/2$ neutral charge state and narrow the spectral distribution of different emitters, facilitating the realisation of entanglement between multiple V centres in telecom-wavelength quantum networks.

## Methods
### Samples
In this work we utilise two SiC samples, labelled as "A" and "B".

Sample A is a ~ 110 $\mu$m thick isotopically-enriched 4H-$^{28}$Si$^{12}$C layer, grown by chemical vapour deposition (CVD) on the Si-face of a standard 4-degrees off-axis (0001) 4H-SiC substrate. The isotope purity is estimated to be 99.85% for $^{28}$Si and 99.98% for $^{12}$C, which was confirmed by secondary ion mass spectroscopy (SIMS) for one of the wafers in the series. The current-voltage measurements using a mercury probe station shows that the layer is n-type with a free carrier concentration of ~ $6 \times 10^{13}$ cm$^{-3}$, which is close to the concentration of the residual N shallow donor of ~ $3.5 \times 10^{13}$ cm$^{-3}$ as determined from low-temperature photoluminescence. Due to contamination from the susceptor, the concentration of the B shallow acceptor is expected to be in the low $10^{13}$ cm$^{-3}$ range. Deep level transient spectroscopy (DLTS) measurements show that the dominant electron trap in the layer is related to the carbon vacancy $V_C$ with a concentration in the range of low $10^{13}$ cm$^{-3}$. In this material, the N shallow donor can compensate the shallow B acceptor and the acceptor levels of $V_C$ to pin the Fermi level at the N shallow donor level.

Sample B is high-purity semi-insulating (HPSI) 4H-SiC material from Cree, used for stabilizing the charge state of single $V^{4+}$ emitters. In this HPSI material, the residual N donors and B acceptors have comparable concentrations, in the low $10^{15}$ cm$^{-3}$ range. The Fermi level is located at the $(0|-)$ acceptor level of the divacancy at ~ $V_B + 2.05$ eV if the concentration of the N donor is larger than that of the B acceptor, or at the $(+|0)$ donor level of $V_C$ at ~ $E_V + 1.75$ eV if B has a higher concentration and can compensate the N donor (see Fig. 4a). In both cases, the neutral charge state $V^{4+}$ is stable since the Fermi level lies between the $(+|0)$ and $(0|-)$ levels of V. In the former case, electrons trapped at the $(0|-)$ acceptor level of the divacancy ($V_{Si} V_C$) may still have weak influence on the PL decay of single $V^{4+}$ emitters.

In both samples, vanadium ions were implanted at an energy of 100 keV, corresponding to a depth of about 60 nm. An implantation dose of $10^8$ cm$^{-2}$ creates a concentration sufficiently low to enable single-emitter studies. To repair the lattice damage created by the implantation process, the sample was annealed at 1400 °C in Ar atmosphere for 30 minutes. The annealing was performed without a C-cap layer and no noticeable degradation of the surface morphology was observed.

### Optical measurements
All measurements were performed with the sample at 4.3 K, mounted in a closed-cycle cryostat (Montana Cryostation s100) with external shroud customised to bring an external neodymium permanent magnet at a distance down to 20 mm from the sample. Optical measurements were performed a with a home-made confocal microscopy setup described in detail in Supplementary Note 1. We use two laser excitations, a narrowband tunable telecom laser (in the range 1270 nm - 1350 nm), in resonance with the ZPL of the V $\alpha$-line[34] and a green repump laser (520 nm) to compensate for laser-induced ionisation[35]. A series of long pass filters in the detection path, allow us to filter out the excitation laser and collect only the phonon sideband of the emission (1300 nm - 1600 nm), which is finally detected by a superconducting nanowire single-photon detector (Single Quantum EOS, with detection efficiency 85% along one linear polarisation). See Supplementary Note 1 for a detailed description of the setup and the experimental measurements.

### Automated detection of quantum emitters
In order to retrieve spectra for each of the spots, we took a sequence of $N$ maps at different detunings $\{f_i\}, i = 1..N$. PL spots associated are detected in each PLE maps by convolving the image with a Gaussian function of a similar width to the confocal spot size and detecting local maxima. The number of photon counts in each spot is then computed integrating the signal over a square as large as the confocal width, for each spot $k$. Spots with centres closer to each other than the spot radius are merged. For each spot $k$, we record the total number of photon counts in each of the maps at different detunings, retrieving the spectrum $P_k(f_i)$ and fit it with a linear combination of Gaussian functions. The centres of all the Gaussians are then used in the histograms in Fig. 3c.

## Data availability
The raw experimental data generated in this study have been deposited in a Zenodo database[69].

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

## Acknowledgements

We thank Caspar van der Wal, Carmem Gilardoni, Roland Nagy, Margherita Mazzera, Erik Gauger, Yoann Altmann and Brian Gerardot for helpful discussions and comments on our manuscript. This work is supported by the European Commission project QuanTELCO (grant agreement No 862721; M.T., C. Bonato, G.B., N.T.S.), the Engineering and Physical Sciences Research Council (EP/S000550/1; C. Bonato), the Leverhulme Trust (RPG-2019-388; C. Bonato), the Austrian Research Promotion Agency project QSense4Power (FFG 877615; M. T.), the Swedish Research Council (VR:2020-05444; J.U.H.) and the Knut and Alice Wallenberg Foundation (KAW 2018.0071; J.U.H, N.T. S.).

## Author contributions

C. Bonato and M.T. conceived the experiments and supervised the project. P.C., M.J.A., C. Bekker, D.W. and C. Bonato constructed the experimental setup. P. C. and M.J.A. performed the experimental measurements. J.U.H designed and performed epitaxial growth, J.U.H and M.G. performed annealing. B.T. and G.B. simulated the expected vanadium spectra and provided general theoretical input relevant for data analysis. P.C., M.J.A., B.T., N.T.S., I.G.I., T.A., P.K. M.T. and C. Bonato analysed the data with input from all co-authors. P.C., M.J.A., B.T., N.T.S., M.T. and C. Bonato prepared the manuscript, with input from all co-authors.

## Competing interests

The authors declare no competing interests.
