## [Peer Review File · Nature Communications]

REVIEWER COMMENTS

Reviewer #1 (Remarks to the Author):

Ultra-narrow inhomogeneous spectral distribution of telecom-wavelength vanadium centres in isotopically-enriched silicon carbide
by Cilibrizzi et al.

Cilibrizzi et al. investigate V⁴⁺ in 4H-SiC using spectroscopy measurements on single emission centres and ensembles as well as measurements with and without magnetic field. Vanadium centres are prospective building blocks to enable future scalable quantum network architectures by providing spin-to-photon quantum nodes/interfaces. A spin-photon interface combines the memory (using single spins) with information transmission capabilities (photons). Hence, the demonstrated spin-dependent optical transitions are particularly important.

Important properties, which the authors study, are the reduction of spectral inhomogeneous broadening as well as the spectral stability of the emitter. A narrow spectral broadening, a small linewidth as well as a good spectral stability are pivotal for providing indistinguishable photons, suitable for single photon nodes in future quantum networks.

By engineering the isotopic composition, the inhomogeneous spectral distribution could be reduced to 100 MHz, which according to the authors is, significantly narrower than any other single quantum emitter to date. They further study the doping concentration and show how this impacts the emission stability. Here, the emission wavelength of Vanadium is within the minimum dispersion O-band which is beneficial for telecommunications.

In general, the work is of great importance and interest for the broader community in quantum communication and sensing. However, at its current state major questions remain which the authors should address prior publication.

The authors structure their manuscript along and abstract/introduction section, three main findings and a discussion section. Please find my comments to each section below.

Abstract & Introduction:

1. “We perform spectroscopy on single emitters and report the first observation of spin-dependent optical transitions [..]”

The authors may clarify if this sentence is a general statement or refers to vanadium specifically.

2. “[..] and allows interfacing with extremely low loss photonic interfaces [2, 30].”

Could the authors elaborate on how a 4H-SiC platform would be interfaced to a fibre or a waveguide in a low loss and scalable fashion such that it meets the loss requirements of photonic quantum networks? Do the authors imagine on chip integrated or external sources/nodes and if integrated, how would such a platform be integrated/interfaced with components such as detectors etc. which are predominantly based on different material platforms?

3. “This distribution is significantly smaller than any other single quantum emitter previously reported in the literature.”

Please provide references in the main manuscript to compare it to the best state of the art single quantum emitters.

Spin-selective optical transitions:

4. “We use a green repump laser (520 nm) to compensate for laser-induced ionization [28].”

Minor: Could the authors elaborate on this. How can the ionization be compensated?

5. Could the authors comment on the large error bars/fluctuations in the $g(2)$ measurement (Fig 2b). What is their main source?

6. “Remarkably, the linewidth of the resonance decreases for higher applied magnetic field.”

The authors may comment on the reason behind this finding. Further, the calculations rely on experimental literature values. Could the authors comment on how comparable/close these samples were to the samples used in this study?

Ultra-narrow inhomogeneous spectral distribution:

7. "Linewidth fitting has given a shift of per unit mass for nearest 22 ± 3 GHz neighbour carbon isotopes and per unit 2.0 ± 0.5 GHz mass for silicon isotopes."

Could the authors comment on where this is shown. Are they referring to a publication or to their own measurement?

8. "For this experiment, we do not confirm the single nature of the emitters with $g(2)$ measurements."

Could the authors comment on how the sample used for this measurement is different than the one used for Fig. 1? Was there a reason for not confirming its single photon emitter nature?

It reads "confirm the single nature of ..." this should be rewritten to "confirm the single photon emitter nature of..."

9. Could the authors comment on potential other effects which could lead to a spectral shift as well as lifetime reduction, such as sample strain due to heating cycles (as mentioned) but also density effects (e.g. up-conversion) and if this can also affect the lifetime or other spectral properties.

10. The authors picked three regions on sample A and showed that each of them exhibits a spectral distribution of about 100 MHz only. How does the choice of the regional size and centre position affect this estimate and why were they chosen this way? Not dividing them into the regions would lead to a much larger overall standard deviation for instance especially when taking region C, which shown in Fig. S10, into account (but also without).

11. Could the authors comment on where the spectral shift of the mean in Fig. S10 between the regions is coming from? And why does the number of emitters vary so strongly from e.g. between region A1 and region A3 although the scan area and the density is constant?

12. "Theoretically the effect of the stress on the spin..."

Please provide a reference.

Stability of the neutral charge state:

13. "The V4+ lifetime of the charge state, which has not been studied before,.."

Minor: Following Nature communications guidelines the authors should generally avoid such claims.

14. "[...] featuring two different doping concentrations"

The authors may want to consider to be more clear which doping is actually meant, as several materials are involved (V, N, B, C ..)

15. The authors compare sample A and B with different doping densities.

It should be made more clear that these are the same samples as previously used.

Moreover, it appears that while sample A (isotopically-enriched) exhibits a narrower spectral distribution, it does now show a shorter lifetime than sample B, i.e. is less stable. Can the authors comment on if one sample can combine both preferential features (i.e. narrow spectral distribution and stable emission).

16. "These experimental observations result from the Fermi level engineering in sample B, which led to the stabilisation of the V4+ state."

Here and in general throughout the manuscript, the authors should make clear if they state an hypothesis or draw a conclusion or refer to knowledge from the literature (with reference).

17. Within this part of the experiment, two doping levels are compared. This is referred to as Fermi level engineering. Could the authors comment if they have optimized the doping, i.e. the Fermi level, and collected more than two data points to achieve a more robust experimental confirmation of their hypothesis?

Discussion:

18. "In this work, we have clarified..."

The authors may consider to write "discussed" instead of "clarified" as although their experimental findings support their hypothesis and conclusion, alternative explanations might be possible too.

19. "... the narrow bands observed for any single emitter"

When making such a claim, the authors should consider listing the recent state of the art results for other emitters to prove this point.

20. "... collective emission experiments"

Could the authors comment/be more specific which collective emission experiments are meant ?

21. "Finally, we engineer the material doping level to ..."

The authors should mention/describe the result of this part of the experiment briefly, e.g. within a following sentence.

22. "... recent work on rare ions has shown that Purcell factors exceeding 500 can be achieved both by photonic crystals and open microcavities."

You may want to consider adding plasmonic waveguides as an additional class Purcell enhancing platforms. See e.g. <https://doi.org/10.1038/s41467-023-38262-6>

23. What is the reason for using different PL map window sizes in Extended Data 1 and 2 i.e. for Sample A vs. Sample B?

As a minor remark, the manuscript could have been composed with some more care. Several grammatical/spelling/typesetting errors exist and although mentioned in the caption, axis labels for some graphs are missing. To mention a few of those mistakes:

- References are wrongly cited/formatted at times
- Line 8: "...associated associated..."
- polarised vs polarized
- quanching vs quenching
- etc.

Reviewer #2 (Remarks to the Author):

The manuscript of Cilibrizzi et al. demonstrated an approach to obtain the ultra-narrow inhomogeneous spectral distribution of vanadium centers in silicon carbide, whose wavelength locates in the telecom range and is beneficial for constructing quantum networking. They found that the inhomogeneous spectral distribution of different vanadium emitters in isotopically-enriched silicon carbide could reduce to 100 MHz, among the most stable single quantum emitters. They further analyzed the stability of the V⁴⁺ charge state by measuring their lifetimes. The experiments and descriptions are systematic and well-presented. This work is timely and essential. I recommend publication after the below concerns are well addressed.

1. about the linewidth of the Kramers doublets. The linewidth of the vanadium ion with spin 1/2 in SiC is relatively larger than silicon vacancy in SiC. The authors have discussed possible reasons for the measured linewidth. Including the comparison of other Kramers spins in SiC would be helpful.
2. about the counts of the single emitter. The authors used a superconducting nanowire single-photon detector to detect the PL counts. They obtain 150 cps for a single emitter during the resonant excitation. It would be helpful to provide the detection efficiency at the phonon sideband wavelength for further collection efficiency improvement.
3. about the measurement sequence. The authors implemented many resonant measurements. It would be better to provide the details of the measurement sequence, for example, the excitation power and the integration time of each data point.
4. about the summary of the inhomogeneous distribution of different emitters. Since divacancy defects in silicon carbide are also critical, including them in the comparison would be helpful.
5. about the typo. There are two "associated" in the introduction; "A sub-set of the maps is shown in Fig. 3(a)" on page 4; the authors should check the text carefully.

Reviewer #3 (Remarks to the Author):

This manuscript studies optical-spin properties of V in isotopically-enriched SiC. Spin defects in wide bandgap semiconductors can be applied to quantum technologies. Authors observed spin-dependent optical transitions under magnetic fields as well as narrow distribution of PLE peaks. These findings shown in this manuscript have enough values for science community (not only quantum technologies but also material science). In addition, the results were obtained by reliable/careful measurements. So, I would say that this manuscript has enough quality for an article of nature communications. I would suggest followings to improve this manuscript before acceptance for publication.

[1] Authors show nice $g(2)$ measurement result (Fig. 1 (b)). I would suggest that authors show the results obtained from fitting such as time constant.

[2] For Fig. 4 (b), (c), please describe more details. Thus, although authors mention the mechanism in supplementary section Note 2, there is not enough explanation in the main text, especially for fig. 4 (b). Please explain about the reasons/mechanism to understand Fig. 4 (b) and (c).

Also, authors mention that pinning of the Fermi level for the sample A occurs due to V_c , but this does not happen in the samples B. How have authors confirmed this?

Thus, if the compensation occurs in the samples B and residual V_c exist in certain amounts, the pinning happens in the samples B too?

[3] I found some typo such as two times “associated” (7th line, left side of page 1), “T2” (“2” should be subscript) (4th line, left side of page 2) and “hown” should be “shown” (6th line from the bottom, left side of page 4). So, please correct.

Reply to reviewers' comments

Referee-1

We thank the reviewer for their very detailed comments on our manuscript, and for stating that *"the work is of great importance and interest for the broader community in quantum communication and sensing"*. We have addressed their comments and suggestions as follows:

1. *"We perform spectroscopy on single emitters and report the first observation of spin-dependent optical transitions [..]"* The authors may clarify if this sentence is a general statement or refers to vanadium specifically.

The sentence refers to vanadium specifically. To clarify this, we modified it to *"We perform spectroscopy on single vanadium emitters and report the first observation of their spin-dependent optical transitions"*.

2. *"[..] and allows interfacing with extremely low loss photonic interfaces [2, 30]."* Could the authors elaborate on how a 4H-SiC platform would be interfaced to a fibre or a waveguide in a low loss and scalable fashion such that it meets the loss requirements of photonic quantum networks? Do the authors imagine on chip integrated or external sources/nodes and if integrated, how would such a platform be integrated/interfaced with components such as detectors etc. which are predominantly based on different material platforms?

While SiC photonics is still in its infancy, SiC features extremely good optical properties (large transparency range, large electro-optic coefficients, etc). A direct path to fiber coupling is provided by the use of microcavities such as those in Ref. 30 (J. Fait et al., Appl. Phys. Lett. 119, 221112 (2021). Supported by recent results related to the fabrication of high-quality optical components, we can definitely envision the development of a more elaborate, all-SiC integrated electro-optic platform including quantum emitters and photonic circuitry (directional couplers, electrically-controlled modulators, tunable filters), etc. This photonic circuitry could then be interfaced to an external fiber through a tapering approach that can reach extremely high efficiencies [see e.g. T. G. Tiecke et al, "Efficient fiber-optical interface for nanophotonic devices", Optica 2, 270 (2015)] and directed to external detectors. Alternatively, some research groups are trying to directly deposit superconducting detectors onto SiC photonic circuitry (e.g. this CLEO paper by F. Martini et al: CLEO, OSA Technical Digest (Optica Publishing Group, 2020), paper FF3D.7.

3. *"This distribution is significantly smaller than any other single quantum emitter previously reported in the literature."* Please provide references in the main manuscript to compare it to the best state of the art single quantum emitters.

All numbers are reported in Table S3 in the Supplementary Information, with relevant literature cited. As *Nature Communication* allows a larger number of references than was permitted for our original *Nature Materials* submission, we have moved the corresponding references from the Supplementary Information to the main text ("*[21, 23, 41–47]*").

4. *"We use a green repump laser (520 nm) to compensate for laser-induced ionization [28]."* Minor: Could the authors elaborate on this. How can the ionization be compensated?

We have elaborated on the need of a repump laser. The quoted sentence is changed to:

"A green repump laser (520 nm) is used to counteract the ionization of V^{4+} (bright state) into V^{3+} (dark state). The necessity of a repump laser is discussed in more detail in the section concerning the stability of the neutral charge and in Supplementary Note 2."

We have also attempted to make the text clearer at the end of the penultimate paragraph of the section "Stability of the neutral charge state":

"The re-pump laser induces excitation of electrons from the acceptor levels of $V(0|-)$ to the conduction band, thus enabling charge state conversion from V^{3+} (dark) to V^{4+} (bright). The decay that we observe in Fig. 4(c) can be explained as an effect of the resonant laser used to probe V^{4+} leading to transformation of V^{4+} into V^{3+} . Indeed, the telecom resonant excitations pumps electrons from

the shallow N donors and V_C acceptor levels into the conduction band, which can be re-captured by V^{4+} to create V^{3+} , leading to the measured exponential decays of its PL on a 129 ms timescale. We notice that the resonant excitation alone is most likely insufficient to reactivate V^{4+} , which stipulates the need of re-pump laser of higher energy (see Supplementary note 2B)”

Finally, we add some more details concerning the interplay between V^{4+} and V^{3+} in a new Supplementary Note, 2B (not pasted here for brevity).

5. *Could the authors comment on the large error bars/fluctuations in the $g(2)$ measurement (Fig 2b). What is their main source?*

The noise in the $g(2)$ measurement is the shot noise on the number of coincidence counts. As we collect only about 150 counts/s from a single vanadium centre, the number of coincidences is relatively low even for integration times as long as several days (as in Fig 2(b)).

6. *“Remarkably, the linewidth of the resonance decreases for higher applied magnetic field.” The authors may comment on the reason behind this finding. Further, the calculations rely on experimental literature values. Could the authors comment on how comparable/close these samples were to the samples used in this study?*

The reason for this is that the off-diagonal hyperfine interaction, which is vastly different for the two involved Kramer doublets, is suppressed by parallel magnetic fields. If the diagonal hyperfine interaction has the same sign for both Kramer doublets, our model and literature parameters (interaction strength) predict similar hyperfine splittings for both Kramer doublets so that the lines narrow when the magnetic field increases.

As long as the defects are in the bulk, i.e. much deeper than the size of their wavefunctions, and no strong (symmetry breaking) strain is involved, we expect the parameters to be sample independent. This is in good agreement with the literature and our measurement, see for instance Fig. 2.

We have added the following explanation to the main text: *“In simple terms, the decrease in linewidth with increasing magnetic field can be understood from the fact that the off-diagonal coupling, which is of a vastly different form for the involved KDs, is suppressed under large magnetic fields and, if the zz components share the same sign, the diagonal coupling leads to a similar hyperfine splitting. Therefore, the model (using the literature values for the hyperfine parameters of “bulk” defects much deeper than the size of their wavefunctions [54], with intact symmetry) predicts that the electron-spin-conserving transitions will converge at higher magnetic fields, resulting in a narrowing of the linewidth.”*

7. *“Linewidth fitting has given a shift of per unit mass for nearest 22 ± 3 GHz neighbour carbon isotopes and per unit 2.0 ± 0.5 GHz mass for silicon isotopes.” Could the authors comment on where this is shown. Are they referring to a publication or to their own measurement?*

These values originate from the fitting of ensemble photoluminescence data from Wolfowitz et al, cited in the main text as Ref [35]. A direct reference is added to the specific sentence.

8. *“For this experiment, we do not confirm the single nature of the emitters with $g(2)$ measurements.” Could the authors comment on how the sample used for this measurement is different than the one used for Fig. 1? Was there a reason for not confirming its single photon emitter nature? It reads “confirm the single nature of ...” this should be rewritten to “confirm the single photon emitter nature of...”*

This measurement is taken on Sample A, as the data in Fig. 1.

The reason for not checking the single emitter nature of the spot in Fig. 2 is that $g(2)$ measurements are extremely time-consuming due to the low count rate, as described in point (5). For these specific measurements, in any case, the single-emitter nature of the photoluminescence spot is not essential (given that the inhomogeneous broadening is much smaller than the linewidth broadened by spectral diffusion).

We have changed “confirm the single nature of ...” to “confirm the single photon emitter nature of...”, as advised by the reviewer.

9. Could the authors comment on potential other effects which could lead to a spectral shift as well as lifetime reduction, such as sample strain due to heating cycles (as mentioned) but also density effects (e.g. up-conversion) and if this can also affect the lifetime or other spectral properties.

The density of vanadium dopants here is sufficiently low to enable single centre studies, i.e. with V centres on average being further from each other than the optical diffraction limit (about 1 centre/ μm^2). In this limit, we do not expect interactions and we believe there should be no impact of density effects on the observed shifts.

Our current best guess is that differences in strain across the sample (for example due to the CVD growth, sample mounting, etc) provide the most likely explanation for the observed shifts between the different regions. Some of the co-authors of this manuscript are currently working on modelling the effect of strain on the properties of transition metal impurities in SiC (more details in the reply to question (12), which we hope will help to clarify the matter.

10. The authors picked three regions on sample A and showed that each of them exhibits a spectral distribution of about 100 MHz only. How does the choice of the regional size and centre position affect this estimate and why were they chosen this way? Not dividing them into the regions would lead to a much larger overall standard deviation for instance especially when taking region C, which shown in Fig. S10, into account (but also without).

Spectroscopy was performed on defects in different regions, separated by more than the scan range of the positioning system, to explore whether the central frequency varies across the sample (for example as a consequence of strain variation across the large scale of the sample). We performed a limited number of scans due to the long acquisition time for each region. Our experiments show that, within one contiguous region, the frequency of almost all emitters is within a very narrow distribution. Additionally, the frequency shift between Regions A1, A2, and A3 is also within a factor of two of the distribution within one region (see below), but Region A4 presents a larger shift. While we do not yet know the origin of this shift, the measurements clearly indicate that multiple samples processed with an identical procedure should contain many regions with emitters at near-identical frequencies, which is the prerequisite for quantum networking. The four regions were picked randomly within the sample, to acquire some indication of how the central frequency may vary across the sample. For practical integrated quantum photonics applications, one would pick areas on each sample with the same frequency.

11. Could the authors comment on where the spectral shift of the mean in Fig. S10 between the regions is coming from? And why does the number of emitters vary so strongly from e.g. between region A1 and region A3 although the scan area and the density is constant?

The spectral shift between the regions A1 and A2 is on the order of 0.2 GHz, which is less than a factor of two greater than the inhomogeneous broadening within each region. The spectral shift between the regions A2 and A3 is far smaller than this value. Our current hypothesis is that the spectral shift of the mean for regions A1, A2, A3 might be related to strain variations over a large spatial scale.

Such material differences may also lead to different incorporation dynamics, affecting the yield in different regions. However, the number of detected spots (27, 33, 37, and 48) is within $\pm 1.69\sigma$ of the mean (38.75). For a normal distribution, such a spread has a likelihood of around 8%. Furthermore, the spot counting algorithm does not identify single emitters, but only spatially resolved peaks, leading to random under-counting and thus greater variability. This effect is clearly seen in the plots for Sample B, where each identified spot contains between one and four resonance peaks, which we interpret as corresponding to multiple emitters. Taking these effects together, it is not very unlikely that the spot count would vary within this range.

12. "Theoretically the effect of the stress on the spin..." Please provide a reference.

In a few words, we can combine the assignment to irreducible representations in *P. Udvarhely et al, Phys Rev B 98, 075201* (ref [57]) and the form of the coupling to electric fields *B. Tissot et al, Phys Rev B 103, 064106* (ref [50]) to model the form of the coupling to strain. This is possible because the model by B. Tissot et al. [50] applies the group theory on the orbital level, and electric field and

strain are both time-reversal symmetric and their components can be assigned to the same irreducible representations of C_{3v} .

To make this clearer, we rewrote the paragraph as follows: *"Theoretically, we will treat the coupling to strain analogously to the coupling to electric fields (see [50]). This is possible for several reasons: the symmetry arguments were used on the orbital level, the fact that the strain coupling fulfils time-reversal symmetry and the possibility to assign (combined) strain tensor elements to the same irreducible representations as the electric field components [57]. In particular, there are two strain tensor elements (i.e., ϵ_{zz} and $\epsilon_{xx} + \epsilon_{yy}$) that transform like an electric field in the z-direction within the C_{3v} symmetry which can directly influence the energy spacing of the KDs."*

We note here that some of the co-authors are finalising a manuscript discussing a detailed theoretical model for the strain interaction based on group theory and density functional theory calculations (B. Tissot, P. Udvarhelyi, A. Gali, and G. Burkard titled "Strain Engineering for Transition Metal Defects in SiC"). This manuscript should be posted on arxiv over the next few weeks.

13. *"The $V4+$ lifetime of the charge state, which has not been studied before,.."* Minor: Following Nature communications guidelines the authors should generally avoid such claims.

We have removed the statement about the $V4+$ lifetime not having been studied before.

14. *"[...] featuring two different doping concentrations"* The authors may want to consider to be more clear which doping is actually meant, as several materials are involved (V, N, B, C ..)

We modified the paragraph to *"performed on samples A and B, featuring two different nitrogen and boron doping concentrations (described in details in Methods).*

15. The authors compare sample A and B with different doping densities. It should be made more clear that these are the same samples as previously used. Moreover, it appears that while sample A (isotopically-enriched) exhibits a narrower spectral distribution, it does now show a shorter lifetime than sample B, i.e. is less stable. Can the authors comment on if one sample can combine both preferential features (i.e. narrow spectral distribution and stable emission).

Samples A and B differ in both their doping and their isotopic composition. Doping is related to adding impurities (i.e. elements other than Si and C) to tune the Fermi level and thereby the stability of specific vanadium charge states. Isotopic composition is related to controlling the specific isotopes of Si (^{28}Si , ^{29}Si , ^{30}Si ,) and C (^{12}C , ^{13}C ,). Therefore these two parameters are completely independent of each other and can be tuned in any combination. We believe it is therefore completely feasible to create an isotopically-enriched semi-insulating sample that would feature both charge state stability and narrow inhomogeneous distribution at the same time.

We have added the sentence *"Doping level and isotopic composition are independent parameters, that can be combined to simultaneously achieve stabilisation of the correct charge state and narrow inhomogeneous distribution on the same sample."* in the Discussion.

16. *"These experimental observations result from the Fermi level engineering in sample B, which led to the stabilisation of the $V4+$ state."* Here and in general throughout the manuscript, the authors should make clear if they state an hypothesis or draw a conclusion or refer to knowledge from the literature (with reference).

Our phrasing *"Fermi level engineering"* can be misleading, as we have not performed a systematic study of the influence of the Fermi level position on the charge state stability. We therefore rephrase it as

"These experimental observations can be understood as due to the different Fermi level in sample B compared to sample A, which results in improved stability of the V^{4+} charge state in sample B."

17. Within this part of the experiment, two doping levels are compared. This is referred to as Fermi level engineering. Could the authors comment if they have optimized the doping, i.e. the Fermi level, and collected more than two data points to achieve a more robust experimental confirmation of their hypothesis?

The main goal of the experiments presented in the section is to show that the lifetime of the V^{4+} charge state is sufficiently long for quantum technology applications. As this goal was achieved in Sample B, we have only examined the two doping levels presented in the main text. A measurement of charge-state lifetime as a function of doping would be interesting from the point of view of materials science, but it has not been conducted and is beyond the scope of the present manuscript.

18. *"In this work, we have clarified..."* The authors may consider to write "discussed" instead of "clarified" as although their experimental findings support their hypothesis and conclusion, alternative explanations might be possible too.

The paragraph has been modified according to the reviewer's suggestions. We have used "investigated", rather than "discussed" or "clarified".

19. *"... the narrowness observed for any single emitter"* When making such a claim, the authors should consider listing the recent state of the art results for other emitters to proof this point.

A comparison between different emitters is reported in Table S3 in the Supplementary Information. We have modified the sentence to "Remarkably, we observe an ultra-narrow 100 MHz spectral inhomogeneous distribution: the comparison in Table S3 in the Supplementary Information highlights that this is the narrowest observed for any single solid-state emitter."

20. *"... collective emission experiments"* Could the authors comment/be more specific which collective emission experiments are meant?

By "collective emission experiments", we mean situations with multiple indistinguishable quantum emitters where "which-path" information is erased, giving rise to bunching in intensity correlation measurements. As this is outside the scope of the paper, we do not further explain it in the main text but give references [62] and [63] for the interested reader to follow up.

21. *"Finally, we engineer the material doping level to ..."* The authors should mention/describe the result of this part of the experiment briefly, e.g. within a following sentence.

We have made this statement more precise by modifying it to: "Finally, we compare the stability of the V^{4+} charge state, featuring telecom emission and a spin $S = 1/2$ state, in high-purity and semi-insulating material. We show that, while in high-purity material the V^{4+} only survives about 100 ms, its lifetime is enhanced by at least two orders of magnitude in semi-insulating silicon carbide."

22. *"... recent work on rare ions has shown that Purcell factors exceeding 500 can be achieved both by photonic crystals and open microcavities."* You may want to consider adding plasmonic waveguides as an additional class Purcell enhancing platforms. See e.g. <https://doi.org/10.1038/s41467-023-38262-6>

We thank the reviewer for the reference, which has been added. We modified the paragraph to "Purcell factors exceeding a few hundred can be achieved by utilising photonic crystals [26], open microcavities [27] or plasmonic waveguides [66]"

23. What is the reason for using different PL map window sizes in Extended Data 1 and 2 i.e. for Sample A vs. Sample B?

We tried to get a similar number of spectrally-resolved spots for samples A and B, to have comparable statistics.

As in the isotopically-enriched sample A the inhomogeneous distribution is much narrower than the PL linewidth, many of the PL spots visible in the maps on sample A are not single emitters but correspond to multiple emitters, all with indistinguishable spectra. In other words, most of the emitters appear as spots in the same map at one given laser detuning frequency. This is also confirmed by the fact that it was not trivial to find spots with $g^{(2)}(0) < 0.5$ in Sample A.

In Sample B, with inhomogeneous distribution much larger than the linewidth, on the other hand, emitters will likely appear in different maps corresponding to different laser detuning frequencies. Given the same density of vanadium emitters, the spot detection algorithm finds here many more spots, as they are distinguishable in frequency.

This analysis shows that, to keep the number of spots checked similar, one needs to examine a larger area on Sample A.

24. As a minor remark, the manuscript could have been composed with some more care. Several grammatical/spelling/typesetting errors exist and although mentioned in the caption, axis labels for some graphs are missing.

We went carefully through the manuscript and corrected the typographical errors we could find, as suggested by the reviewers.

Referee-2

We thank the reviewer for defining our work as "*timely and essential*", and recommending publication. We have addressed their comments and suggestions as follows:

1. about the linewidth of the Kramers doublets. The linewidth of the vanadium ion with spin 1/2 in SiC is relatively larger than silicon vacancy in SiC. The authors have discussed possible reasons for the measured linewidth. Including the comparison of other Kramers spins in SiC would be helpful.

The linewidths observed here are larger than those seen in group IV defects in diamond, which have many strong similarities to this system. We cannot state the origin of this relative broadening conclusively. While the Kramers doublets are insensitive to lowest order to strain and electric fields, the crystal field term of the optical transition is sensitive to charge fluctuations in the vicinity of the defect. The magnitude of this effect is currently being studied, but we hope that the referees will agree that it is beyond the scope of this work.

2. about the counts of the single emitter. The authors used a superconducting nanowire single-photon detector to detect the PL counts. They obtain 150 cps for a single emitter during the resonant excitation. It would be helpful to provide the detection efficiency at the phonon sideband wavelength for further collection efficiency improvement.

Our detectors have a nominal efficiency of 85%. They do have a strong polarisation dependence, with 85% efficiency for one polarisation and much lower for the orthogonal one.

We updated the corresponding sentence in Methods as "*...which is finally detected by a superconducting nanowire single-photon detector (Single Quantum EOS, detection efficiency 85% along one linear polarisation)*"

3. about the measurement sequence. The authors implemented many resonant measurements. It would be better to provide the details of the measurement sequence, for example, the excitation power and the integration time of each data point.

We agree with the reviewer that this information is important for improving the reproducibility of the research. We have now modified the manuscript by adding the excitation powers and integration times of the resonant excitation measurements both in the figure captions and in the main text.

In particular in the main text we have added:

"We identify V^{4+} centres by confocal spectroscopy at 4.3 K (see Methods and Supplementary Note 1), using a narrowband tunable CW laser (1278.8 nm, with typical excitation powers ranging from 1 μ W to 4 μ W depending on the specific experiments) to resonantly excite the α zero-phonon line (ZPL) of V^{4+} [29], while detecting the phonon sideband emission 1300 nm - 1600 nm). We use a green repump laser 520 nm, 14 μ W) to compensate for laser-induced ionization [30]"

4. about the summary of the inhomogeneous distribution of different emitters. Since divacancy defects in silicon carbide are also critical, including them in the comparison would be helpful.

We agree with the reviewer that the divacancy in SiC is a very promising spin defect. We prefer not to list it on Table S3 as we are not aware of any experimental value for its inhomogeneous broadening. Given its C3v electronic structure and S=1 electronic spin, we however expect it to be very similar to the NV centre in diamond. A comment has been added to the Supplementary Information in Supplementary Note 4A: "*We did not include the divacancy in SiC in the Table, as we could not*

find an experimental value for its inhomogeneous broadening in the literature: however we expect its performance to be similar to the NV centre in diamond, as it possesses an identical electronic structure and spin state."

5. about the typo. There are two "associated" in the introduction; "A sub-set of the maps is shown in Fig. 3(a)" on page 4; the authors should check the text carefully.

We went carefully through the manuscript and corrected the typos we could find, as suggested by the reviewers.

Referee-3

We thank the Reviewer for assessing that our findings "have enough values for science community (not only quantum technologies but also material science)" and that "the results were obtained by reliable/careful measurements". We have addressed their comments and suggestions as follows:

1. Authors show nice $g(2)$ measurement result (Fig. 1 (b)). I would suggest that authors show the results obtained from fitting such as time constant.

As indicated by the reviewer, we have added the following sentence into the caption of Fig. 1b: "The yellow line corresponds to a single exponential fit function, $f(x)=1 - A e^{(-|t|/t_1)}$, with $A = 0.828 \pm 0.163$ and $t_1 = 0.048 \pm 0.013 \mu s$."

The reduction of the time constant in the $g(2)$ measurement compared to the optical lifetime is an effect of the strong optical pumping, as shown by I. Aharonovich *et al*, Phys. Rev. A 81, 043813 (2010). We have added a comment about this is in the figure caption ("The decay constant here is different than the known value for the emitter lifetime, as it depends on the excitation optical power [49]"). Please note that the same effect was seen by G. Wolfowitz *et al* (reference [35] in the main text).

2. For Fig. 4 (b), (c), please describe more details. Thus, although authors mention the mechanism in supplementary section Note 2, there is not enough explanation in the main text, especially for fig. 4 (b). Please explain about the reasons/mechanism to understand Fig. 4 (b) and (c). Also, authors mention that pinning of the Fermi level for the sample A occurs due to V_C , but this does not happen in the samples B. How have authors confirmed this? Thus, if the compensation occurs in the samples B and residual V_C exist in certain amounts, the pinning happens in the samples B too?

In addition to the new section on charge state dynamics added as Supplementary Note 2B in the Supplementary Information (see response to question 16 of referee 1), we have added the following paragraph to the main text to comment on both part (b) and (c) of Fig 4, as suggested:

"Furthermore, the higher the resonant laser power, the higher the excitation rate for free electrons, leading to higher free-electron concentration and increased capture rate to the V^{4+} centres. This explains the observed decrease in the decay time observed in Fig. 4(b)".

The pinning of the Fermi level in sample A is due to the nitrogen shallow donor. This is stated in several points in the main text (e.g.in Methods: "In this material, the N shallow donor can compensate the shallow B acceptor and the acceptor levels of V_C to pin the Fermi level at the N shallow donor level", on page 5: "Specifically, in sample A, the Fermi level is pinned to the donor level of N", etc). In one occasion, on page 5, we were stating that "the telecom resonant excitations pumps electrons from the V_C acceptor levels into the conduction band, which can be re-captured by V^{4+} , leading to the measured exponential decays of its PL on a 129 ms timescale". We corrected this to "Indeed, the telecom resonant excitations pumps electrons from the shallow N donors and V_C acceptor levels into the conduction band, which can be..."

3. I found some typo such as two times "associated" (7th line, left side of page 1), "T2" ("2" should be subscript) (4th line, left side of page 2) and "hown" should be "shown" (6th line from the bottom, left side of page 4). So, please correct.

We went carefully through the manuscript and corrected the typos we could find, as suggested by the reviewers.

REVIEWERS' COMMENTS

Reviewer #1 (Remarks to the Author):

The authors have addressed my comments sufficiently well and I am happy to recommend the work for publication in Nature Communications. As mentioned, it would be certainly interesting if the authors could elaborate on the effect of strain in future works.

Reviewer #2 (Remarks to the Author):

The authors have addressed all my concerns. I recommend publishing it in Nature Communications.

Reviewer #3 (Remarks to the Author):

This manuscript studies optical-spin properties of V in isotopically-enriched SiC. Spin defects in wide bandgap semiconductors can be applied to quantum technologies. Authors observed spin-dependent optical transitions under magnetic fields as well as narrow distribution of PLE peaks. These findings shown in this manuscript have enough values for science community (not only quantum technologies but also material science). In addition, the results were obtained by reliable/careful measurements. So, I would say that this manuscript has enough quality for an article of nature communications. Also, the answers to my comments are reasonable. So, I would suggest the revised version of this manuscript can be accepted for publication.